# European Mistletoe (*Viscum album*) Extract Is Cytotoxic to Canine High-Grade Astrocytoma Cells In Vitro and Has Additive Effects with Mebendazole

**DOI:** 10.3390/vetsci9010031

**Published:** 2022-01-15

**Authors:** Anna Wright, Rie Watanabe, Jey W. Koehler

**Affiliations:** 1Auburn University Honors College, Cater Hall, 277 Thach Concourse, Auburn, AL 36849, USA; acr1821@jagmail.southalabama.edu; 2Department of Pathobiology, College of Veterinary Medicine, Auburn University, 166 Greene Hall, Auburn, AL 36849, USA; rzw0041@auburn.edu

**Keywords:** canine, cancer, dog, glioma, glioblastoma, mistletoe, *Viscum album*, mebendazole

## Abstract

Malignant gliomas are associated with extremely poor clinical outcomes in both humans and dogs, and novel therapies are needed. Glioma-bearing canine patients may serve as promising preclinical models for human therapies, including complementary medicine. The objective of this study was to evaluate the effects of mistletoe extract (*Viscum album*) alone and in combination with mebendazole in an in vitro model of canine high-grade astrocytoma using the cell line SDT-3G. SDT-3G cells were exposed to a range of concentrations of mistletoe extract alone to obtain an IC50. In separate experiments, cells were exposed to mebendazole at a previously determined IC50 (0.03 µM) alone or in conjunction with varying concentrations of mistletoe extract to determine the additive effects. The IC50 for mistletoe alone was 5.644 ± 0.09 SD μg/mL. The addition of mistletoe at 5 μg/mL to mebendazole at 0.03 µM led to increased cell death compared to what would be expected for each drug separately. The cytotoxicity of mistletoe in vitro and its additive effect with mebendazole support future expanded in vitro and in vivo studies in dogs and supply early evidence that this may be a useful adjunct therapeutic agent for use in glioma-bearing dogs. To the authors’ knowledge, this is the first published report of *Viscum album* extract in canine glioma.

## 1. Introduction

Malignant gliomas represent a significant proportion of both human and canine primary brain tumors and have a similarly poor prognosis despite treatment in both species [1,2,3,4,5,6]. Chemotherapeutic treatment of malignant gliomas is often thwarted by features such as the blood–brain barrier, the presence of resistant stem-like cell subpopulations, and hypoxic microenvironments [7,8,9]. Canine malignant gliomas may be promising models for their human counterparts in many aspects, and there is a growing body of literature that documents the validity of this model and increasing interest [10,11,12,13,14,15]. Because the classification system of canine gliomas has recently been updated [16] and the molecular landscape of canine gliomas has yet to be fully elucidated, “high-grade astrocytoma” or the more broad “malignant glioma” when appropriate will be used in this manuscript in lieu of glioblastoma when referring to canine tumors. 

The traditional standard of care for treating glioblastoma, the most common and malignant form of glioma in humans, includes surgery followed by radiation and chemotherapy [17]. The most commonly used chemotherapy is temozolomide (TMZ); however, its cytotoxicity is inhibited in O-6-methylguanine-DNA methyltransferase (MGMT)-expressing tumors. Of human glioblastomas, 65% lack promoter methylation-induced silencing of *MGMT* and, therefore, do not respond optimally to TMZ treatment [18,19,20]. Very little is known about the *MGMT* promoter methylation status of canine malignant gliomas, but preliminary epigenetic investigations into a small subset of tumors suggests no evidence of significant *MGMT* promoter methylation in those tumors (Dr. Kevin Woolard, personal communication). Additionally, little is known about the efficacy of TMZ in canine glioma patients [21,22], although work in the Dickinson lab and in our lab (unpublished data) suggests in vitro IC50 in canine malignant glioma cell lines that are well outside achievable in vivo concentrations and are similar to or above those of resistant human tumors [3,23]. Therefore, while TMZ may be fairly well tolerated in dogs [24] and is effective at increasing survival in a subset of human brain tumor patients, it is clear that other effective treatments are urgently needed for the many human and canine patients who may not respond to TMZ.

A promising alternative or adjunct treatment undergoing investigation in treating malignant gliomas is mebendazole (MBZ), a benzimidazole (BZ) anthelmintic that has been used successfully for many years to treat intracranial and intestinal helminth infections. The anthelmintic effects of BZ are primarily mediated by binding to tubulin and preventing its polymerization [25], and this mechanism also imparts antineoplastic properties to the drug. Prevention of tubulin polymerization interferes with assembly of the mitotic spindle, arresting it in the G_2_-M phase of the cell cycle and ultimately triggering apoptosis [26]. In its original investigation of use in gliomas, MBZ was found to preferentially target neoplastic cells, to slow tumor growth in vitro, and to prolong survival in xenograft mouse models using both human xenografts and syngeneic mouse tumors [27]. Since its discovery as a potential antineoplastic therapy, the BZ has been shown to have cytotoxic effects on lung, breast, and ovary cancer cells as well as colon carcinomas, osteosarcomas, and gliomas [26,28]. MBZ is lipophilic, allowing it to cross the blood–brain barrier and reach the tissue at significant levels [28]. However, MBZ’s hydrophobic nature can also be a limiting property, providing a very limited level of absorption when taken orally [29]. Oral absorption can be enhanced for many BZs by administration with a fatty meal, and the drug is tolerated very well even at high treatment doses [28,29]. Our lab has shown that MBZ has good efficacy against multiple canine malignant glioma cell lines in vitro at IC50 concentrations that should be biologically achievable [30]. 

Interest in European mistletoe (*Viscum album*) extract for use in oncologic complementary medicine has grown in the last decades, especially in its indigenous European countries, due to its ability to repeatedly demonstrate significant cytotoxicity in many different human tumor cell lines, including leukemia, lymphoma, breast cancer, glioma, and others in vitro [31,32,33,34,35,36]. Whole-plant mistletoe extract contains primary active ingredients consisting of glycoproteins or mistletoe lectins (ML) and less-potent polypeptides (viscotoxins) as well as secondary ingredients without any direct effect on immunology such as alkaloids, polysaccharides, and amino acids [37]. ML-1, the most predominant lectin, comprises two subunits. The 34-kDa B-chain binds carbohydrates and is responsible for their apposition to receptors on target cell membranes [38]. The 29-kDa A-chain subunit inactivates 60S ribosomes by the same mechanism as ricin’s A-chain, which is by cleaving the N-glycosidic bond of A-4324 of 28S rRNA, rendering the ribosome unable to undergo protein synthesis and, thus, activating the mitochondrial-dependent pathway for apoptosis [39,40]. In the human leukemic cell line MOLT-4, ML-1 causes apoptosis when used at low or intermediate concentrations, while exposure to high concentrations at long intervals causes cell death primarily through necrosis [41]. When treated with ML-1 in vitro, monocytes/macrophages and lymphocytes are activated, allowing the release of significant levels of IL-1a, IFN-y, and TNF-a, cytokines that are capable of stimulating T cells, natural killer cells, and macrophages that attack and kill tumor cells [42,43,44,45,46,47]. The extract also has antiangiogenic properties, which limit tumor proliferation and metastasis [48,49]. In intracerebral syngeneic rat models using the F98 cell line, mistletoe treatment produces a diminution of gliomas as well as a prolonged relapse-free survival [35,36]. Mistletoe extract also reduces glioblastoma cell motility in human and murine models in vitro, and it reduces tumor growth in vivo in human xenograft mouse models treated intratumorally or subcutaneously; it also reduces tumor growth in syngeneic mouse glioma models when treated intratumorally [50]. 

*Viscum album* is a hemiparasitic plant that grows in a variety of host trees, and extracts from the plant vary in chemical composition based on the species of the host tree on which they grow [51,52]. For this project, we chose to evaluate the in vitro effects of European mistletoe from the basswood host tree on the canine high-grade astrocytoma cell line SDT-3G. This specific extract was selected due to its superior efficacy as compared to others in in vitro experiments using human medulloblastoma cell lines [52]. There are currently no studies examining mistletoe in canine cancer patients in the English literature. In published German language-only studies of *Viscum album* use in canine cancer patients [5,6], there were statistically significant survival benefits in canine patients with oral melanoma treated with radiation and *Viscum album* with doses ranging from 0.1 to 20 mg subcutaneously three times per week for the entire survival time (up to 4 years); survival in canine mammary carcinoma patients treated with surgery and mistletoe vs. surgery alone was not statistically significantly improved but showed a trend toward improvement. In the Italian language-only literature there is a study evaluating the use of mistletoe in dogs with transmissible venereal tumor that showed a reduction in vincristine dose in dogs being treated with *Viscum album* [53]. In order to evaluate whether mistletoe might merit further study into its clinical benefit in canine malignant glioma patients either alone or as adjunct therapy, we evaluated the effects of the drug by itself as well as in conjunction with MBZ. 

## 2. Materials and Methods

### 2.1. Cell Culture

The canine high-grade astrocytoma cell line SDT-3G (a generous gift from Dr. Pete Dickinson) was originally classified as a grade IV astrocytoma/glioblastoma and was derived from a spontaneous primary tumor in a 12-year-old male English Bulldog. This cell line has copy number alterations in EGFR (gain), AKT (gain), PIK3CA (gain), CDKN2A/B (homozygous loss), and NF1 (homozygous loss) and lacks TP53 mutations [54]. SDT-3G cells were cultured in standard 15-mL polystyrene tissue culture flasks in Dulbecco’s Modified Eagle Medium (DMEM) with 4.5 g/L glucose, glutamate, and sodium pyruvate, supplemented with 10% heat-inactivated fetal bovine serum (FBS) (“standard media”), in a 37 °C humidified incubator with 5% CO_2_. Prior to experiments, subconfluent cells were trypsinized and counted manually using the trypan blue method. Then, 3500 cells/well were seeded into 96-well plates and allowed to attach for approximately 18 h prior to the addition of test drugs.

### 2.2. Viscum album and Mebendazole Preparations

*Viscum album* extract (*Viscum album* Iscucin^®^ Tiliae, Strength “H”, WALA Heilmittel GmbH, Eckwälden, Germany) was provided as a gift from the manufacturer. The standardized Strength H is a 5% solution of the entire plant extract, containing 50 mg/mL of *Viscum album* in an aqueous solution of water, NaCl, and NaHCO_3_. This equates to 13.7 μg/mL of the lectin active ingredient [52]. When diluted to preparations for cell viability assays, the 1-mL ampule was broken and any extract not used for that experiment was put into a vacuum-sealed tube via syringe and stored at room temperature in the dark, per the company’s instructions. Extract was then diluted to desired experimental concentrations in standard media. Dilutions were made fresh for each experiment. Cells were exposed to a range of concentrations from 5 μg/mL to 0.1 pg/mL for 72 h, followed by analysis of cell viability using the MTT assay. 

In the experiments evaluating additive effects of mistletoe extract and MBZ, one “high” concentration of mistletoe (“HMT”), 5 μg/mL, and one “low” concentration (“LMT”), 2.5 ng/mL, were separately supplemented to the MBZ at IC50 concentration of 0.03 μM, for 72 h, followed by analysis of cell viability using the MTT assay. 

### 2.3. MTT Cell Viability Assays

The MTT (3-(4,5-dimethylthiazol-2-yl)-2,5-diphenyltetrazolium bromide) cell viability assay provides an estimate of the number of metabolically active cells; viable cells have the ability to convert MTT to a formazan product that produces a colorimetric reaction detected by absorbance around 570 nm. Following exposure of cells to drugs for 72 h, MTT reagent was added to experimental and control wells and allowed to incubate for 4 h at 37 °C. Then, cells were lysed and solubilized by the addition of DMSO. Absorbance from the colorimetric reaction was read in an automated plate reader (Infinite^®^ M200 plate reader, Tecan Group Limited, Männendorf, Switzerland) at 570 nm with a reference of 630 nm subtracted for background effect [55]. The choice of a 72-h exposure time, while arbitrary, is a common interval in published cytotoxicity experiments and, therefore, allows for comparison across studies. All experiments were performed three times on three separate days. In each experiment, all test wells were performed in technical quadruplicate; all experiments included media-only well controls and no-treatment well controls. Raw optical density (OD) values generated by the plate reader were further analyzed as follows: Media-only absorbance values were averaged and subtracted from experimental and no-treatment control values to account for any absorbance caused by the supporting media. No-treatment control values were then averaged, and both experimental and control groups were expressed as percentages of the average control value. 

### 2.4. Statistical Analysis

Data were analyzed using statistical software (Prism© Version 7.02, GraphPad Software, Inc., La Jolla, CA, USA), with log-transformation of drug concentrations followed by generation of IC50 values using a four-parameter, variable-slope, curve fit, nonlinear regression analysis with automatic outlier exclusion. Comparison of effects across groups was measured by ANOVA with statistical significance set at *p* ≤ 0.05, using the Brown–Forsythe test for equal variances. 

## 3. Results

Exposure of the SDT-3G canine high-grade astrocytoma cell line to mistletoe extract alone for 72 h generated an average IC50 of 5.644 ± 0.09 SD μg/mL (Figure 1). The generated data suggested that the cytotoxicity of *Viscum album* on this cell line is dose-dependent, has a low standard deviation among technical replicates in each experiment, and is highly repeatable across experiments performed on different days at different cell passages. 

Exposure of the canine SDT-3G canine malignant glioma cell line to MBZ at its IC50 (0.03 μM) for 72 h alone and in conjunction with either a high (HMT, 5 μg/mL) mistletoe concentration or low (LMT, 2.5 ng/mL concentration was evaluated. The “high” dose was based on the IC50 value of *Viscum album* cytotoxicity experiments. The “low” dose was based on an in vitro study showing significant apoptosis after administration of ML-1 purified mistletoe lectin with a concentration of 2.5 ng/mL to human leukemic cells [38]. 

Cells exposed to MBZ at its IC50 showed the expected approximately 50% decrease in cell viability as compared to untreated cells (55.5% live cells vs. control, *p* < 0.0001), while the addition of MBZ with HMT generated a significant further decrease in cell viability as compared to MBZ-only treated cells (39.3% greater cell death, *p* = 0.0034) or mistletoe-only treated cells at the same dose (40% greater cell death, *p* = 0.0008) (Figure 2). Treatment with LMT combined with MBZ had no impact on cell death beyond the effect attributable to MBZ (*p* = 0.98). 

## 4. Discussion

The initial study of the cytotoxicity of *Viscum album* alone indicated that mistletoe extract does have a dose-dependent cytotoxic effect on the canine SDT-3G malignant glioma cell line in vitro. In pharmacokinetic studies, the C_max_ refers to the peak serum concentration that a drug achieves after administration of a single dose, while the T_max_ (or area under the curve) is the amount of time the drug is at the C_max_ in serum. Although there is little primary pharmacokinetic data for mistletoe, in one study in which healthy human volunteers were given a single subcutaneous administration of the product containing 20 μg of natural lectins, there was considerable interindividual variability, with a plasma C_max_ range of 188 to 2970 pg/mL and a T_max_ range of 0.3 to 336 h [56]. There is little direct pharmacokinetic experimental evidence of mistletoe’s ability to cross the blood–brain barrier, but there is secondary evidence of penetration via its efficacy in animal seizure models [57], rodent cerebral xenograft and syngeneic models [50], and human glioma patients [36]. The blood–brain barrier is frequently disrupted in gliomas, which theoretically should lead to increased drug exposure of tumor cells, at least in some regions of the tumor. In addition, besides systemic administration, possibilities for its use in glioma patients could be by intratumoral administration with or without incorporation into biopolymers or similar substances [58,59]. 

There are no published pharmacokinetic data on mistletoe in dogs and almost no published safety data; however, one toxicology study on recombinant ML-1 showed no genotoxic or mutagenic effects at doses up to 1000 ng/kg intravenously, and repeated subcutaneous or intravenous dosing of rats and dogs up to 1000 ng/kg did not produce any specific target-organ toxicity. In dogs administered 10,000 ng/kg intravenously daily for 1 week, there were bleeding complications; local reactions at the sites of subcutaneous injections occurred at doses greater than 50 ng/mL [60]. 

## 5. Conclusions

This work represents the first published data regarding the efficacy of mistletoe in canine malignant glioma. While these initial results seem promising, the study has limitations and many questions remain regarding the safety and efficacy of mistletoe in tumor-bearing dogs and people. The IC50 data from the present study, in conjunction with other published experimental data, suggest that alternatives to systemic delivery of whole-extract products (such as localized tumor therapy or isolation and concentration of the bioactive lectins) may be necessary to achieve therapeutically efficacious levels for direct cytotoxicity. However, patients may still benefit from the immunomodulatory or additive effects even at sub-IC50 levels, but that will require appropriate experimental investigation in clinical trials in glioma-bearing dogs. Additionally, in vitro investigation in multiple canine glioma cell lines and in vitro and in vivo evaluation of mistletoe’s additive effects with multiple other chemotherapeutic combinations used in canine glioma such as temozolomide and lomustine would provide valuable information. With regard to the possibility of local administration, it is interesting to note that in one study, syngeneic mouse models that carried mammary carcinoma subcutaneously in both hind legs experienced more mistletoe-induced cytotoxicity in the tumor nearest the flank injection site than in the contralateral tumor or metastatic tumors, suggesting some direct regional effects in addition to the systemic ones [61]. Treatment with mistletoe in human patients in the clinical setting has been associated with mild but not limiting side effects, including headache, fever, chills, and soreness and inflammation at the injection site [62,63,64]. Dogs in the previously mentioned oral melanoma studies occasionally had similar transient side effects at the highest doses [6]. Less common but more serious effects in humans include circulatory issues, swelling of lymph nodes, allergic reactions, and increased intracranial pressure, which could be a limiting factor in treating glioma patients [65]. There are many chemical components in this complex plant extract that could be responsible for both therapeutic efficacy and adverse events, including lectins, amines, acetylcholine, choline, histamine, alkaloids, tyramine, and acids [37]. Additionally, the composition of these components varies depending on the tree species on which it is grown [37]. A limitation of the present study is the use of a single cell line and the evaluation of drug efficacy in cultured cells arranged in a monolayer in nutritive media and 20% oxygen. Another limitation of this cell culture drug cytotoxicity model is the inability to account for the significant immunomodulatory effects of this drug, which would require in vivo testing in immunocompetent animal models. Clearly, standard cell culture conditions do not accurately recapitulate the in vivo environment; nevertheless, studies like these provide a useful starting point for further investigations into chemotherapeutic agents of interest and are not commonly performed for complementary medicine modalities. Overall, the significant in vitro cytotoxicity of mistletoe to canine SDT-3G glioma cells as both a sole and additive treatment with MBZ, the demonstrated low side effects in tumor-bearing canine patients [5,6], and its status as a drug of significant interest in human complementary medicine make it a compelling candidate for future pharmacokinetic, safety, and efficacy studies in dogs with malignant gliomas [66,67,68,69,70]. 

## Figures and Tables

**Figure 1 vetsci-09-00031-f001:**
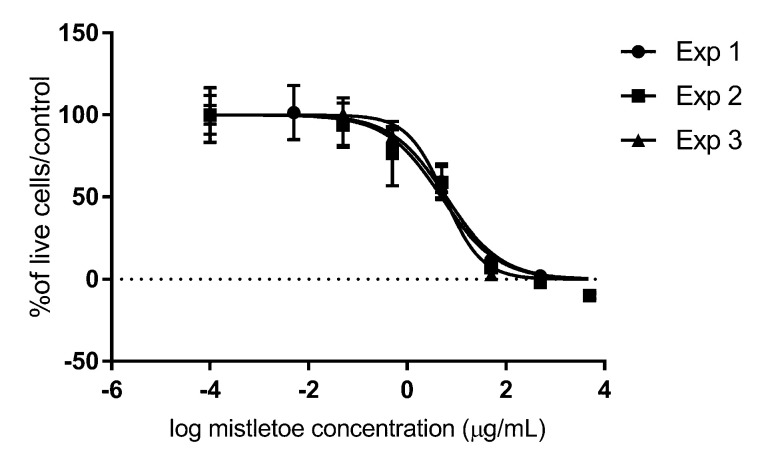
In vitro cytotoxic effects of *Viscum album* Iscucin^®^ Tiliae on canine SDT-3G glioma cells. Cells were exposed to a range of concentrations of mistletoe extract (5 μg/mL, 2.5 ng/mL, 1 ng/mL, 1 pg/mL, and 0.1 pg/mL) and incubated for 72 h before cell viability was evaluated with MTT assays, which is represented as a percentage of viable cells in the treatment group as compared to untreated control cells. “Exp 1, 2, 3”means experiment 1, 2, 3.

**Figure 2 vetsci-09-00031-f002:**
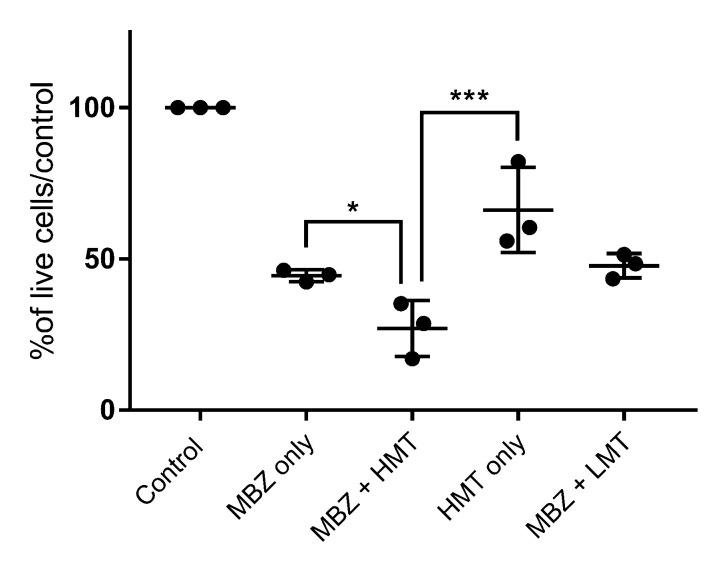
In vitro impact of MBZ alone and in combination with *Viscum album* on canine SDT-3G glioma cells. Cells were incubated for 72 h with either MBZ (0.03 μM) only, mistletoe (5 μg/mL) only, or combined 0.03 μM MBZ + high mistletoe (“HMT”, 5 μg/mL) or MBZ + low mistletoe (“LMT”, 2.5 ng/mL) before cell viability was evaluated with MTT assays. Data are shown as a percentage of viable cells in the treatment group as compared to untreated control cells, with each dot representing the mean of technical replicates for a given experiment. * is statistically significant (*p*
*≤* 0.05); *** is highly significant (*p* < 0.001).

## Data Availability

Raw data are available upon request.

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
