# Peer review of "European Mistletoe (Viscum album) Extract Is Cytotoxic to Canine High-Grade Astrocytoma Cells In Vitro and Has Additive Effects with Mebendazole"

_vetsci, 2022, doi:10.3390/vetsci9010031_

Round 1

Reviewer 1 Report

This study evaluated cytotoxicity of Viscum album extract and its additive effect with mebendazole in the canine high grade astrocytoma cell line SDT-3G. Initially, the IC50 for mistletoe was determined, then increased cell death was observed at the addition of mebendazole with the IC50 of mistletoe. Because effective medications are not available for canine malignant gliomas yet, this preliminary data will be a basis for the clinical application of mistletoe extract in combination with mebendazole in dogs. I have very few comments to make in regard to the overall quality of the study.

  1. In the Introduction, third paragraph is relatively long. To enhance the understanding of readers, I recommend to divide it into two paragraphs. The beginning of new paragraph could be “Interest in European mistletoe 75 (Viscum album) extract for use in oncologic complementary medicine ~”.
  2. In this experiment, high grade astrocytoma cell line was exposed to mistletoe extract and mebendazole. If their effects were examined for low grade astrocytoma, please add the information.
  3. Did you compare the cytotoxicity between chemotherapeutic agent (such as temozolomide and lomustine) and mistletoe or mebendazole? If you have the result or related experiments exist, please describe it.
  4. MTT assays: Cells were exposed to drugs for 72 hours. Please explain the reason of setting an exposure time.

Author Response

Thank you very much for your careful review and helpful comments.  The introduction has been divided for readability.  A low grade astrocytoma was not tested, nor was there any testing of mistletoe with other chemotherapy drugs (a fantastic idea); the scope of this study was pre-submitted to the editors to gauge interest and they indicated that it was of interest as-is.  An explanation of the 72 hour interval was added.

Reviewer 2 Report

The authors obtained promising results with regard to the influence of the mixture of mistletoe extract & mebendazole. However, they used only one cell line and I would suggest adding more canine tumor cell lines to the research. Furthermore, they used solely MTT viability test, that is a simple method for evaluating cell viability. I would recommend adding some assays detecting apoptotic cells as well (flow cytometry maybe). In the present form the research it is difficult to discuss, because there are few conclusions to draw. In the discussion section the Authors made a lot of statements that are based on other literature,

for example 231-233 The Authors claim that ‘The data from the present study, in conjunction with other published experimental data, suggests that localized tumor therapy’ ‘ I do not think that data gained from cell culture may give evidence of or suggest localized tumor therapy. This conclusion can be drawn only from other publications.

I suggest publishing the paper as short communication if possible, or after adding some assays by the authors.

Additionaly, there are some statements missing which are usually required, e.g.

lack of statement concerning conflict of interest, financial statement etc.

There are also some language issues. Please check carefully throughout the text and standardize the spelling, and correct the typos.

These are only some examples:

16-17 Abstract IC50 and IC50 –please unify

170 was measured. The sentence lacks a verb.

174 canine SDT-3G canine high grade astrocytoma cell line – is it necessary to repeat the word ‘canine’?

212 with a plasma Cmax range of 188 to 2970 pg/mL and a tmax range of 0.3 to 212 336 – what are C max and t max – it needs to be further explained

214 the authors use either blood brain barier without a hyphen or blood-brain barier with a hyphen. Please unify throughout the text.

The Figure 1 is illegible. I would suggest changing putting lines in the chart in different colors. Maybe this will help.

Author Response

Thank you for your careful review of the manuscript and your helpful comments.  Some additional wording has been added to increase the length to conform to the 3000-word Article format.  Please note that the editors were contacted prior to submission about the scope of the experimental data and they indicated that it was of interest as-is.  Some wording has been altered/removed/added to correct minor unification or grammatical errors.  We appreciate your close attention to detail.  A brief explanation of Cmax and Tmax have been added for clarity.  For figure 1 we will defer to the editor; the main reason for showing the three curves is to show how little inter-experiment variation there is and the slope of the curve, which we believe it does even though the overlap prevents visual separation in a monochromatic figure.  

Round 2

Reviewer 2 Report

Most of the things that I raised have been corrected. I am satisfied with the present version of the manuscript.